# Trick-or-Trap: Extracellular Vesicles and Viral Transmission

**DOI:** 10.3390/vaccines11101532

**Published:** 2023-09-27

**Authors:** Juan-Vicente Bou, Shuhei Taguwa, Yoshiharu Matsuura

**Affiliations:** 1Laboratory of Virus Control, Center for Infectious Disease Education and Research, Osaka University, 2-8 Yamadaoka, Suita, Osaka 565-0871, Japan; 2Research Institute for Microbial Diseases, Osaka University, 3-1 Yamadaoka, Suita, Osaka 565-0871, Japan; 3Center for Advanced Modalities and DDS, Osaka University, 2-8 Yamadaoka, Suita, Osaka 565-0871, Japan

**Keywords:** extracellular vesicles, viral transmission, collective infectious units, cluster transmission, vesicle transmission

## Abstract

Extracellular vesicles (EVs) are lipid membrane-enclosed particles produced by most cells, playing important roles in various biological processes. They have been shown to be involved in antiviral mechanisms such as transporting antiviral molecules, transmitting viral resistance, and participating in antigen presentation. While viral transmission was traditionally thought to occur through independent viral particles, the process of viral infection is complex, with multiple barriers and challenges that viruses must overcome for successful infection. As a result, viruses exploit the intercellular communication pathways of EVs to facilitate cluster transmission, increasing their chances of infecting target cells. Viral vesicle transmission offers two significant advantages. Firstly, it enables the collective transmission of viral genomes, increasing the chances of infection and promoting interactions between viruses in subsequent generations. Secondly, the use of vesicles as vehicles for viral transmission provides protection to viral particles against environmental factors, while also expanding the cell tropism allowing viruses to reach cells in a receptor-independent manner. Understanding the role of EVs in viral transmission is crucial for comprehending virus evolution and developing innovative antiviral strategies, therapeutic interventions, and vaccine approaches.

## 1. Introduction

Viral infection is commonly perceived as a highly efficient process. However, PFU/particle ratios indicate that most viral particles produced by viruses are not infectious [1]. This can be due to genetic defects caused by high mutation rates [2] or structural defects caused by environmental physicochemical conditions if the viral particle takes a long time to find a susceptible cell [3]. However, even when viral particles do not have any defects, the success of infection is not guaranteed. During the early stages of infection, viral components face barriers to infection that may cause their degradation and lead to abortive infections, such as lysosome or proteasome degradation in case the virus takes too long to exit from the endosome and/or uncoating the viral particle [3,4,5]. In other words, as soon as the infection process begins, a countdown starts, and the virus must reach critical levels of viral factors, through protein translation and genome replication, which will allow it to overcome these initial barriers. The permissiveness of the cell to be infected will determine the probability of a successful infection since this affects the virus’s replication rate, the number of viral components required, or the time the virus has to produce them before the infection is aborted [3]. Therefore, the infection process is not a walk in the park, and the virus must overcome numerous obstacles to achieve successful infection.

One way to increase their chances in the infectious process is to initiate infection with a larger number of viral genomes [6]. This can reduce the stochastic effects associated with initiating infection, as well as accelerate the rates of virus replication and translation, which facilitates the virus reaching critical levels of viral components that allow it to overcome these barriers [7]. Under the classic paradigm of “one infectious particle—one viral genome” for initiating infection with a larger number of genomes, it is necessary to achieve a large population size, at least locally, which allows multiple viral particles to reach the same cell [8]. However, our perception of this paradigm has changed, as we now know that viruses can transmit a group of genomes through collective infection units such as: (1) polyploid capsids, viral particles containing multiple copies of their genomes; (2) direct cell-to-cell transmission, through intercellular connections such as plasmodesmata, nerve synapses, immunological synapses or virological synapses; (3) viral aggregates, multiple viral particles associate, directly with each other, or indirectly with extracellular organic matter or on the surface of other cells; (4) occlusion bodies, baculoviruses form polyhedrin crystallized matrices containing multiple viral particles inside; (5) extracellular vesicles (EVs), membranous structures involved in intercellular communication [9]. The latter case is especially interesting since extracellular vesicles are produced by almost any eukaryotic or prokaryotic cell and EV-mediated viral transmission has been described in all types of viruses (Table 2). For this reason, in this review, we will address the role of extracellular vesicles in the process of viral infection and transmission.

## 2. Extracellular Vesicles

Extracellular vesicles are non-replicative particles bounded by a lipid membrane produced by almost any eukaryotic or prokaryotic cell [10] and have been successfully observed in all types of biological samples: blood, urine, feces, semen, cerebrospinal fluid, synovial fluid [11]. Initially, it was believed that they fulfilled the function of disposing of useless cellular components [12]. However, EVs contain bioactive molecules such as proteins, lipids, and nucleic acids that can reach all types of cells and alter their behavior. EVs are now considered one of the most important mechanisms of intercellular communication that play a role in the maintenance of homeostasis and participate in the pathogenesis of cancer, neurodegenerative diseases, infectious diseases, and cardiovascular diseases [13,14,15,16].

The EVs are widely diverse in size (from 30 nm to 10 μm in diameter), biogenesis pathway (multivesicular bodies, secretory autophagy, plasma membrane budding, apoptosis), lipid composition (cholesterol and phospholipids such as ceramide, sphingomyelin, and phosphatidylserine), and membrane protein components (tetraspanins such as CD9, CD63, and CD81, integrins, chaperones, etc.) [11]. This diversity complicates the creation of a single nomenclature system that can adequately categorize all types of EVs that have been described. Due to this, the International Society for Extracellular Vesicles (ISEV) cannot provide a clear criterion for the classification of EVs. Instead, it recommends defining an operational classification in each case. We have chosen to use a nomenclature that allows us to distinguish the different types of EVs that play an important role in viral infection and transmission.

Exosomes (ES) originate from the endocytic pathway of a cell, specifically from multivesicular bodies (MVBs) which are formed by the invagination of the endosomal membrane [17,18]. As MVBs mature, small intraluminal vesicles (ILVs) are generated and packaged within them. These ILVs are then released from the MVBs by fusion with the plasma membrane, forming exosomes that are subsequently secreted into the extracellular medium [19]. Their diameter ranges from 30 to 250 nm.

Microvesicles (MV) are formed by the shedding or budding of membrane-bound vesicles directly from the plasma membrane of cells [20]. Microvesicles can range in size from 100 to 1000 nm in diameter.

Autophagosome-like microvesicles (ALMV) are released from cells through a process called secretory autophagy [21]. Autophagosomes are intracellular vesicles that have a double membrane and are involved in the autophagy pathway [22]. They are responsible for recycling and degrading cytosolic components and can originate from various cell membranes, including the endoplasmic reticulum, Golgi apparatus, mitochondria, or plasma membrane [22,23,24]. While autophagosomes typically migrate to lysosomes for degradation, in secretory autophagy, they can migrate to the plasma membrane to release ALMVs [21,25]. They have a diameter ranging from 100 to 1000 nm.

Apoptotic bodies (AB): When cells undergo apoptosis (programmed cell death), they release large microvesicles called apoptotic bodies that are over 1000 nm in diameter [11]. These vesicles contain cellular debris and are enclosed by a phospholipid layer that presents phosphatidyl serine on their surface. Phosphatidylserine acts as an “eat me” signal that attracts phagocytic cells to clear away the apoptotic bodies and their contents.

## 3. Role of Extracellular Vesicles in Viral Infection

The components of the membrane of the extracellular vesicle can interact with several biomolecules present on the surface of the target cell, such as heparan sulfate, integrins, lectins, and tetraspanins [26,27,28,29]. These interactions can activate the internalization of the vesicle and trigger intracellular signaling cascades that modify the cell’s gene expression. In addition, the vesicle’s content can also alter the behavior of the recipient cell. While certain types of extracellular vesicles, including cytochalasin B-induced membrane vesicles (CIMVs), exhibit a lack of specificity when it comes to fusion with target cells. In these cases, the critical determinant for successful fusion lies in the heterophilic interaction between CIMV membrane receptors and the surface proteins of the target cells [30]. The extracellular vesicles can release their cargo into the cytoplasm in two ways: direct fusion with the plasma membrane or fusion with the endosomal membrane after its internalization through the endocytic pathway [31].

### 3.1. Antiviral Functions of EVs

As mentioned above, EVs mediate intercellular communication between all types of cells, including immune cells. Therefore, it is expected that the host uses EVs to regulate the immune response and combat pathogenic infections [32,33,34] (Table 1).

By transporting cytokines, EVs can mediate the inflammatory response and recruitment of immune cells at the site of infection. For example, macrophages infected with the Dengue virus release exosomes containing the proinflammatory cytokine IL-6, as well as other proteins with immune functions such as complement protein C3 and Galectin-3 [35]. Infected cells can introduce viral components into EVs, such as nucleic acids or viral proteins. In the recipient cell, nucleic acids can be recognized as pathogen-associated molecular patterns (PAMPs) by pattern recognition receptors (PRRs), such as toll-like receptors (TLRs), which activate the production and release of interferon (IFN) and other antiviral cytokines [36]. For example, cells infected with HIV-1 produce exosomes containing viral miRNAs that can induce TNF-α production in macrophages [37]. IFN, in turn, activates more antiviral activity genes called interferon-stimulated genes (ISGs), which can also be transferred by exosomes [38]. Antigen-presenting cells (APCs) can capture EVs carrying viral proteins, process these proteins, and use them for antigen presentation [39,40], giving EVs an indirect role in antigen presentation. However, they are not limited to the dissemination of antigens. APC-derived EVs produce exosomes containing antigens bound to MHC-I and MHC-II, thereby exercising a direct role in antigen presentation and participating in the activation of CD4+ and CD8+ T lymphocytes and the activation and production of memory cells [41,42,43].

Extracellular vesicles can also transmit cellular components that exhibit antiviral functions. APOBEC3G is a cellular antiviral protein that deaminates cytosines causing G to A mutations [44]. The HIV-1 Vif protein counteracts the action of this antiviral mechanism. However, it has been described that the transmission of APOBEC3G via uninfected cells can combat HIV-1 infection [45]. Furthermore, placental trophoblasts contain cellular miRNAs that confer resistance to infection by certain viruses. In cell culture experiments, it has been demonstrated that these placental cells can transfer their anti-herpes simplex virus 1 (HSV-1) and anti-cytomegalovirus (CMV) resistance to non-placental cells through EVs carrying these miRNAs [46].

There are many similarities between the entry pathways of enveloped viruses and EVs [31]. This might cause receptor competition phenomena between EVs and viral particles that could play a protective antiviral role [47]. Similarly, T lymphocytes produce CD4^+^ exosomes. This tetraspanin is one of the receptors of HIV-1 and its presence on the membrane of exosomes can act as a mechanism for neutralizing circulating viral particles, reducing the effective viral load [48].

### 3.2. Proviral Functions of EVs

EVs play a crucial role in transmitting biological signals between distant cells, and the host can utilize them as a defense mechanism against infections. However, it is important to note that EVs can also serve as vehicles for viruses, allowing them to exploit these vesicles for their own benefit (Table 1).

In addition to transporting biomolecules with antiviral functions such as miRNAs and cytokines [35,46], EVs can facilitate the infection process by transporting biomolecules with proviral functions. For instance, coxsackievirus (CVB) and herpes simplex virus 1 (HSV-1) have been shown to promote the secretion of miRNAs with proviral functions, which enhance the infection of recipient cells [49,50]. Similarly, Newcastle disease virus (NDV) exploits EVs to transport proviral miRNAs that inhibit the interferon-beta (IFN-β) response, aiding in viral replication [51]. In the case of Epstein–Barr virus (EBV) infection, infected cells release EVs, including galectin-9, an immunosuppressive molecule that interacts with TIM-1 to induce apoptosis of CD4+ T lymphocytes. These EVs also negatively regulate the activation of macrophages and T lymphocytes, impairing the immune response [52]. Moreover, EBV-infected B cells can secrete LMP1 through EVs, a viral protein that plays a crucial role in the transformation of B lymphocytes while inhibiting the proliferation of T lymphocytes and the cytotoxicity of NK cells [53,54]. The cells infected with HIV-1 can secrete Nef, a viral protein with important proviral functions: it can modulate lipid rafts by facilitating the fusion of viral particles [55]. It also reduces the incorporation of CD4 into exosomes, thereby evading this defense mechanism and promoting the spread of infection [48]. Furthermore, Nef negatively regulates the production of IgG and IgA in B lymphocytes [56]; There is also a belief that EVs containing Nef can reactivate latent HIV-1 infection [57].

In addition to utilizing EVs to transport components with proviral functions, viruses employ other mechanisms to facilitate their transmission. One such mechanism is the promotion of viral receptor transfer from infected to uninfected cells with HIV-1, enhancing the susceptibility of recipient cells to infection [58,59]. Similarly, ACE2-containing EVs promote the infection of SARS-CoV-2 [60,61]. Human herpesvirus 6 (HHV6) blocks the antigen presentation in infected cells by packaging major histocompatibility complexes in viral particles and exosomes [62]. Human T cell leukemia virus type 1 (HTLV-1) utilizes EVs to transport viral proteins and RNA, which stimulate the formation of cell–cell contacts [63].

### 3.3. A New Infectious Particle: EVs as Vehicles for Viral Transmission

In 2003, the retrovirus infection models of the time, which assumed that viral infection requires interaction between the envelope glycoprotein and the cellular receptor, were unable to explain certain observations that had occurred in HIV and other retrovirus research. Notably, HIV retained 1% of its infectivity even when deprived of its envelope protein, suggesting the existence of an alternative infection pathway [64,65]. Stephen Jay Gould and colleagues proposed the Trojan exosome hypothesis, according to which HIV and probably other retroviruses could take advantage of the exosome biogenesis pathway to transmit themselves silently and independently of the env–CD4 interaction [66]. The idea of a new infectious particle was born at this time, the extracellular vesicles.

Several years later, one study demonstrated that HIV-1 can be captured and endocytosed by dendritic cells, translocate through the endosomal pathway to multivesicular bodies without de novo synthesis, and be released back in exosomes that can infect CD4^+^ T lymphocytes, a process that the authors coined as trans-infection and that reinforced the Trojan exosome hypothesis [67]. Over time, it has been shown that many other viruses, in addition to retroviruses, can exploit the biogenesis pathways of extracellular vesicles for cell–cell, tissue–tissue, and host–host transmission. The severe fever with thrombocytopenia syndrome virus (SFTS virus) can infect and spread to new cells by incorporating one to five viral particles within the extracellular vesicle [68]. The hepatitis C virus and Flaviviruses such as Dengue and Zika viruses can also be transmitted via EVs by the incorporation of infectious viral genomes in exosomes [69,70,71,72,73]. Marseilleviruses are DNA viruses that infect amoebas and can produce infectious microvesicles with viral particles inside [74]. Other enveloped viruses, such as JEV, yellow fever virus, and VSV, can incorporate their nucleocapsids in ILVs [75,76]. Although EV has not been described in these viruses, this seems to be quite plausible.

Non-enveloped viruses have been considered to require cell lysis for the release of their progeny. However, in 2013 it was demonstrated that the hepatitis A virus is capable of using the exosome biogenesis pathway to form enveloped viral particles that can be released without the need for lysis of the cell [77]. Later, the same was shown to be true for the hepatitis E virus [78,79]. Enteroviruses, another example of non-enveloped viruses, can exploit the secretory autophagy pathway to release viral particles in autophagosome-like microvesicles [80,81,82]. In addition to enteroviruses, other picornaviruses such as the encephalomyocarditis virus, a cardiovirus, can introduce viral particles into both autophagosome-like microvesicles and exosomes [83]. Rotavirus and norovirus can also be transmitted in groups within microvesicles and exosomes, respectively [84]. JC polyomavirus can secrete viral particles in exosomes. The transmission of the JC virus through exosomes could explain how it can infect cells that lack the necessary receptors for virion fusion [85,86].

Indeed, the employment of extracellular vesicles as vehicles for viral transmission is a highly pervasive phenomenon within the virosphere (Table 2). Consequently, it is reasonable to infer that comprehending this transmission mechanism is crucial for gaining insights into viral evolution and pathogenesis.

## 4. Pros and Cons of Viral Transmission through Extracellular Vesicles

The transmission of viruses through extracellular vesicles should have significant implications in the evolution of viruses, given their widespread presence in the virosphere. The evolutionary implications of vesicle transmission on viral fitness can be categorized into the effects of increased multiplicity of infection (MOI) by spreading multiple viral particles or genomes in a single infection unit and the effects of using extracellular vesicles as an infection route. Notice that the classical concept of MOI represents the population-level relationship between the number of viruses and cells; in this case, we omit uninfected cells and focus only on infected cells, assigning value to the average number of viral genomes that initiate infection.

### 4.1. Effects Derived from an Increase in MOI

Transmission through extracellular vesicles allows viruses to transmit multiple viral particles or genomes in a single infectious unit. This helps maintain the benefits of a high MOI even if the virus population density is low. It has been suggested that this increase in MOI at the cellular level may have important implications for the biological fitness of viruses [9,87,88] (Figure 1).

Firstly, the most evident effect of clustering viruses within EVs is the dispersal cost (Figure 1A). That is, if an extracellular vesicle contains *n* infectious particles or genomes within it, this vesicle can initiate the infection of a single cell with *n* viral genomes. However, under the monodispersal model, the virus could potentially infect *n* cells. In terms of productivity, this cost can only be compensated if there is a positive correlation between the size of the founding group—the number of infectious genomes initiating the infection of the cell—and productivity—the number of infectious genomes that the infected cell is able to produce—known in ecology as the Allee effect [89]. If the cell infected by a vesicle containing *n* infectious particles produces a progeny >*n* times greater than in monoinfection, this cost will be compensated, and net profit will be obtained. In other words, if the viral genomes collaborate within the infected cell and increase the yield per capita. A recent study demonstrated both experimentally and through mathematical modeling that viral replication is an inherently cooperative process [90]. This cooperative nature of viral replication could compensate for the dispersal cost associated with collective transmission.

Secondly, free virions often fail in the infection process [3]. However, initiating infection with a higher number of viral genomes can produce a mass effect that increases the probability of infection (Figure 1B). A study on the Vaccinia virus observed a disproportionate increase in the probability of infection as the number of genomes entering the cell increases [6]. It has also been shown in several viral species that increasing the MOI can accelerate the viral infection cycle and help it overcome the cell’s antiviral barriers, as well as favor viral replication under adverse conditions [7,74,84,91,92,93,94].

Thirdly, it has been suggested that EVs could improve the biological fitness of viruses through cooperation between viral quasispecies [88]. That is, events of cooperation between different genetic variants of the same virus. For example, the deleterious effects of a mutation in a trans-acting protein—a protein that can be exploited by several viruses under coinfection conditions—can be compensated by genetic complementation [95] (Figure 1C). In the case of beneficial mutations that show negative epistasis effects—negative interaction between the two mutations when present in the same genome—cooperation between different variants over time could favor a division of labor in which each variant specializes in improving one of its functions, resulting in an increase in population-level average fitness [96,97,98] (Figure 1D). However, the simultaneous transference of distinct viral variants can also lead to unfavorable consequences, such as the emergence of selfish mutations or promoting negative dominant effects. Certain mutations may exert a negative dominant influence on the functional variant. This phenomenon has been strategically harnessed in the development of antiviral treatments targeting structural proteins within viruses possessing polymeric capsids. The objective is to minimize the likelihood of resistant variants emerging [99]. When a resistance mutation does occur, the host cell ends up harboring two coexisting variants: the wild-type, which remains susceptible to the antiviral treatment, and the mutant, which has acquired resistance to it. This coexistence results in the production of progeny with mosaic capsids, composed of a blend of sensitive and resistant proteins to the antiviral agent. Intriguingly, the phenotype of these progeny is predominantly influenced by the sensitive variant, regardless of whether the viral particle’s genome contains the resistance mutation or not. Notably, viruses that are transmitted within clusters or groups further impede the emergence of resistance variants in that context, thereby bolstering the effectiveness of dominant drug targets. This same principle applies to the context of neutralizing antibodies (Figure 1E), which could significantly hinder the virus’s ability to evade immune responses induced by vaccines. Moreover, the co-transmission of different variants can also favor the emergence of selfish variants that maximize their own fitness at the expense of the fitness of other variants [100] (Figure 1F). The most extreme case is that of defective interfering particles (DIPs), which have huge deletions and therefore replicate much faster than full variants and can interfere with their functioning [101]. It has been observed that VSV viral aggregates promote the emergence of DIPs and take cover of the viral population in only three serial transfers [102]. However, in a recent study with coxsackievirus b3 (CVB3), it was observed that most of the extracellular vesicles produced by cells coinfected with two variants of this virus contained only one of them [103]. CVB3, like other positive-strand RNA viruses, induces the formation of replicative organelles using intracellular membranes from the endoplasmic reticulum and Golgi apparatus, where replication and virus morphogenesis take place [104,105,106,107,108,109,110]. Therefore, it is likely that there is a spatiotemporal coupling between viral replication and EV incorporation of new viral particles. This could cause the genomes or viral particles within a single vesicle to be sibling variants originating from the same parental genome. Although further studies with other viruses transmitted through extracellular vesicles are needed, this study suggests that the genetic diversity within extracellular vesicles is limited by kin selection to new mutations that have occurred during the last replication cycle, so interactions—positive or negative—based on genetic diversity will be unlikely and short-lived.

### 4.2. Vehicular Effects of Extracellular Vesicles

The use of EVs as a vehicle of transport can also have implications for the process of viral infection and pathogenesis due to the physicochemical characteristics of the vehicle (Figure 2), regardless of the number of viral genomes or particles contained within it. Due to their larger size, extracellular vesicles will disperse more slowly according to the laws of diffusion and may have difficulties penetrating some host physical barriers. A study with CVB3 showed that vesicle transmission spreads slower than classical free viral transmission [111]. The same study also demonstrated that the adsorption rate of EVs is lower than that of free virus, suggesting that vesicle infection takes more time in binding and entry.

Extracellular vesicles can be isolated from all types of biological samples, suggesting that they are highly resistant to environmental conditions such as pH, temperature, humidity, or UV radiation [112] (Figure 2A). The lipid composition of exosome membranes is enriched in cholesterol and sphingolipids, while microvesicles are enriched in ceramide and sphingomyelin [113,114], which may play a role in their resistance to freeze–thaw cycles, treatment with detergents, and chlorine disinfection [112,115]. Therefore, the use of EVs as a vehicle could be a mechanism of protection against environmental conditions, allowing the virus to maintain its infectivity for a longer period. Santiana and colleagues isolated viruses and extracellular vesicles from feces of mice infected with norovirus and rotavirus and observed that the viruses found within exosomes and microvesicles were in their mature and infectious form, while most of the free viral particles were damaged and degraded [84]. Infecting mice with these EVs is more effective than infecting them with the same amount of infectious viral particles. Since the transmission route of these viruses is fecal–oral, the authors suggested that EVs protect the viruses from both the acidic pH of the stomach during their entry and intestinal proteases during their exit, demonstrating that EVs can play a protecting role during host-to-host transmission. Viruses can also use EVs for immune escape, traveling within these vehicles protects the viruses from the presence of surrounding neutralizing antibodies, thereby allowing for more persistent infections [77,83,116,117]. However, the incorporation of viral particles into EVs may also be a cellular defense mechanism. HAV exosomes are more susceptible to being captured by pDCs that appear to have adapted to detect foreign agents transmitted by EVs [118]. Moreover, some enveloped viruses tend to incorporate their glycoproteins in the envelope of EVs, making them susceptible to antibody recognition [119].

The interactions between the structural proteins of viruses and their receptors present on the membranes of susceptible cells are a key factor in determining the tropism of many viruses [120,121]. Extracellular vesicles represent a new infectious unit that may allow viruses to infect cells through alternative pathways that do not involve interaction with their classic viral receptors (Figure 2B). This has been observed with HIV and JC polyomavirus infecting cells independently of their receptor [64,65,86]. Other viruses, such as the SFTS virus, may also incorporate viral particles into EVs to transmit and infect new cells independently of their receptor [68]. Hepatitis C virus transmitted in EVs, and probably any virus that incorporates free infectious genomes in EVs, requires an alternative route to interaction with its receptor for infection [69,70]. It has also been suggested that EV transmission may be helpful for vertical transmission and to facilitate the crossing of placental barriers [122,123]. Thus, vesicular transmission allows viruses to use alternative entry mechanisms that expand the range of susceptible cells to infection.

Finally, phosphatidylserine (PS) is a phospholipid that is highly prevalent in cellular membranes, including the plasma membrane. Healthy cells utilize specific enzymes to maintain this lipid on the intracellular side. During apoptosis, these enzymes cease to function, and PS spontaneously flips to the extracellular side, where it serves as a signaling function to attract phagocytic cells to engulf apoptotic bodies formed by the sick cell [124]. The interaction of PS with its receptors initiates the production and release of anti-inflammatory cytokines such as IL-10 and TGF-β. Many enveloped viruses incorporate PS into their envelope to disguise themselves as apoptotic bodies and trick phagocytic cells into infecting them without alerting the immune system. This process is known as viral apoptosis mimicry [125]. Many EVs also incorporate PS into their membranes, therefore potentially being able to utilize viral apoptosis mimicry to infect cells through PS receptors without alerting the immune system. This may be particularly interesting in the case of non-enveloped viruses. HAV, for example, uses TIM-1, a PS receptor, to infect cells, which makes sense considering that it can be transmitted through exosomes [77]. Enteroviruses may also utilize PS receptors to infect cells when transmitted in autophagosome-type microvesicles [82]. Although enterovirus EVs still require interaction with the classical virus receptor, enteroviruses may benefit from the immune-suppressing effect of PS [126].

## 5. Clinical Applications and Concluding Insights

Extracellular vesicles (EVs) have emerged as remarkable vehicles for viral transmission, offering a host of advantages such as heightened infectivity, promotion of cooperative interactions, protection against environmental conditions and neutralizing antibodies, and an expanded cell tropism (Table 2). The widespread utilization of EVs in the virosphere underscores their significance (Table 1). Indeed, targeting viral transmission within vesicles has proven to be a promising strategy in combating coxsackievirus [127]. However, it is also essential to acknowledge their drawbacks, including the emergence of cheater viruses, increased per unit infection costs, reduced dispersal speed, slower binding, and entry kinetics and bolstering dominant drug targets. These shortcomings shed light on why viruses often opt for a mixed progeny, utilizing both the conventional mechanism of free viral particles and extracellular vesicles for transmission.

Beyond their role as viral transmission vehicles, several studies have demonstrated that extracellular vesicles have significant therapeutic and immunization potential. Despite reports suggesting that EVs may facilitate immune evasion by the virus [77,83,116,117,128], it has also been shown that the immune system is capable of detecting these infectious vesicles and responding appropriately [118,119]. Understanding this process could represent a crucial step in advancing new vaccination strategies. In the case of enveloped viruses, the presence of their structural proteins in the membrane of extracellular vesicles has been observed [71,73,128,129,130,131,132,133,134,135,136,137], making them susceptible to recognition by the immune system as foreign agents. In particular, glycoproteins on envelope virus particles are conformational antigens that are easily recognized by B cells, and for effective vaccination, they must be sensitized with appropriate lipid membranes that retain their intrinsic epitope structure. Kaute et al. replaced the cytoplasmic and transmembrane domains of the SARS-CoV S protein with those from the VSV glycoprotein. In mouse models, vesicles harboring this modified version of the SARS-CoV spike protein demonstrated the ability to elicit immunity on par with adenoviral vaccines, and they spurred a higher production of neutralizing antibodies compared to SARS patients [138]. Another strategy employed in crafting vesicle-based vaccines involves constructing viral-like vesicles housing the replicase of the Semliki Forest virus (SFV) and a viral glycoprotein. This approach has yielded safe and effective vaccines for VSV, SIV, HBV, and the rabies virus [139,140,141,142]. The utilization of the arrestin domain containing 1 (ARRDC1)-mediated microvesicles (ARMMs) serves as an efficient system for antigen presentation. Fusion proteins comprising WW domains, which interact with the PPXY motifs of ARRDC1, along with viral epitopes like Flu M2 or HIV gp41, have demonstrated remarkable effectiveness in antigen presentation, leading to robust protection against lethal virus infections [143]. Surprisingly, a similar achievement has recently been made with CVB3, a non-enveloped virus [144]. In this study, Zhang et al. developed a vaccine based on exosomes isolated from VP1 stably HEK293T cells containing the CVB3 VP1 protein. Remarkably, a dose ten times smaller of this vaccine generated nearly twice as much immunity as prior vaccines containing recombinant VP1.

The prospect of harnessing the unique properties of extracellular vesicles (EVs) has opened up fascinating avenues for the development of new vaccination strategies, providing a promising foundation upon which to build and advance in the fight against infectious diseases, with ongoing efforts to enhance the scalability of EV production for clinical applications [145]. This innovative approach, which focuses on the transmission and utilization of EVs, has the potential to revolutionize the field of vaccine research, potentially leading to safer and more effective immunization strategies against a wide range of viral pathogens. Moreover, EVs hold promise in improving systemic cancer treatment with oncolytic viruses [146]. However, it is crucial to exercise caution, as the misuse of EVs might favor an excessive immune response and cause a massive secretion of proinflammatory cytokines, leading to an uncontrolled inflammation process known as a cytokine storm. Moreover, EVs can also play proviral functions (Table 2), so it is necessary to fully understand the role played by extracellular vesicles in each scenario. Finally, the discovery of transmission through extracellular vesicles has blurred the distinctions between enveloped and non-enveloped viruses. We need to continue investigating the role of extracellular vesicles in the infection process as it may lead to important discoveries that can help us to better understand virus evolution and explore new antiviral strategies.

**Table 2 vaccines-11-01532-t002:** Viral transmission in extracellular vesicles.

	Virus Name	Virus Family	Genome	VP Size	EV Type	EV Size	Infectious Cargo	Markers	References
**Non-enveloped**	Human astrovirus	Astroviridae	ss(+)RNA	30 nm	ES	100–200 nm	VP	CD63, Alix	[147]
	Cricket paralysis virus	Dicistroviridae	ss(+)RNA	30 nm	ES	30–100 nm	NG & VP	Alix, Flotillin-1, Rab35, Syntaxin-1A	[148]
	Hepatitis E virus	Hepeviridae	ss(+)RNA	30 nm	ES	50 nm	VP	CD9, CD63, CD81, Alix, Tsg101	[79,149]
	Hepatitis A virus	Picornaviridae	ss(+)RNA	30 nm	ES	50–110 nm	VP	CD9, CD63, CD81, Alix, Flotillin-1	[77,150]
	Duck hepatitis A virus	Picornaviridae	ss(+)RNA	30 nm	ES	30–150 nm	NG & VP	CD63, Tsg101	[151]
	Poliovirus	Picornaviridae	ss(+)RNA	30 nm	ALMV	300–400 nm	VP	LC3, calnexin, PS	[82,152]
					ES & MV	80, 170 nm	NG & VP	CD9, PS	[153]
	Rhinovirus	Picornaviridae	ss(+)RNA	30 nm	ALMV	300–400 nm	VP	LC3, calnexin, PS	[82]
	Coxsackievirus B1	Picornaviridae	ss(+)RNA	30 nm	NS	100–300 nm	VP	β-actin	[154]
	Coxsackievirus B3	Picornaviridae	ss(+)RNA	30 nm	ALMV	300–400 nm	VP	LC3, calnexin, PS	[82,103]
					ES	100 nm	VP	Alix, CD9	[155]
	Enterovirus A71	Picornaviridae	ss(+)RNA	30 nm	ES	100 nm	NG & VP	CD63, Tsg101	[156]
	Enterovirus D68	Picornaviridae	ss(+)RNA	30 nm	ALMV ?	100–300 nm	VP	NS	[157]
	Echovirus 16	Picornaviridae	ss(+)RNA	30 nm	ES	70–200 nm	VP	CD9, CD63, CD81	[158]
	Encephalomyocarditis virus (EMCV)	Picornaviridae	ss(+)RNA	30 nm	ES & MV	50–350 nm	VP	CD9, Flotillin-1, LC3	[83]
	Foot and mouth disease virus	Picornaviridae	ss(+)RNA	30 nm	ES	<200 nm	NG	CD9, CD63, Alix	[159]
	Norovirus	Caliciviridae	ss(+)RNA	40 nm	ES	<200 nm	VP	CD9, CD63, CD81	[84]
	Infectious bursal disease virus	Birnaviridae	segmented-dsRNA	70 nm	ALMV	500 nm	VP	LC3	[160]
	Rotavirus	Reoviridae	segmented-dsRNA	80 nm	MV	300–500 nm	VP	CD98, PS	[84,115]
					ES & MV	110–450 nm	VP	Alix, CD63, GM1, Integrin-α2	[161]
	Rice gall dwarf virus	Reoviridae	segmented-dsRNA	70 nm	ALMV	200–500 nm	VP	ATG8	[162]
	Avian orthoreoviruses	Reoviridae	segmented-dsRNA	70–80 nm	ES	100 nm	VP	Tsg101, Hsp70	[163]
	Bluetongue virus	Reoviridae	segmented-dsRNA	80 nm	ALMV	300–1000 nm	VP	Annexin A2, LAMP1, LC3, Tsg101	[164]
	Trichomonasvirus	Totiviridae	linear-dsRNA	40 nm	ES	30–150 nm	VP	TvTSP1	[165]
	Torquetenovirus	Anelloviridae	ssDNA	30 nm	ES	70 nm	NG & VP	CD63, CD81, Annexin II	[166]
	JC polyomavirus	Polyomaviridae	circular dsDNA	50 nm	ES	150–200 nm	VP	CD9, CD81, Flotillin-1, Annexin-V, TSG101	[85,86]
	BK polyomavirus	Polyomaviridae	circular dsDNA	50 nm	ES	50–100 nm	VP	CD9, CD63, CD81	[167]
**Enveloped**	Porcine reproductive and respiratory syndrome virus	Arteriviridae	ss(+)RNA	45–60 nm	ES	30–150 nm	NG	CD9, CD63, Alix	[129]
	Porcine epidemic diarrhea virus	Coronaviridae	ss(+)RNA	120 nm	ES	100 nm	NG	CD9, CD63, Alix	[117]
	SARS-CoV-2	Coronaviridae	ss(+)RNA	120 nm	ES	>120 nm	VP	NS	[168]
					AB	1.6–9.5μm	VP	NS	[169]
	Hepatitis C virus	Flaviviridae	ss(+)RNA	50 nm	ES	50–100 nm	NG	CD9, CD63, Alix, Tsg101	[69,70]
	Hepatitis G virus	Flaviviridae	ss(+)RNA	50 nm	ES	NS	NG	NS	[170]
	Dengue virus	Flaviviridae	ss(+)RNA	50 nm	ES	50–150 nm	NG & VP	CD9/AalCD9, CD81/AalCD81	[71,72]
					ALMV ?	2–5μm	NG & VP	LC3, Rab11, Transferrin receptor	[116,171]
	West nile virus	Flaviviridae	ss(+)RNA	50 nm	ES	30–200 nm	NG	CD9	[71]
	Tick-borne Langat virus	Flaviviridae	ss(+)RNA	50 nm	ES	30–200 nm	NG	CD9	[131]
	Zika virus	Flaviviridae	ss(+)RNA	50 nm	ES	50–150 nm	NG	CD9, CD63, Alix	[51,132]
					ES & MV	125, 320 nm	NG	CD63, PS	[172]
					NS	300–700 nm	VP	NS	[173]
	Chikungunya virus	Togaviridae	ss(+)RNA	70 nm	ES	50–250 nm	NG	CD9, CD63	[133]
	Human immunodeficiency virus	Retroviridae	ss(+)RNA - RT	80–100 nm	ES	100 nm	VP	CD1b, CD9, CD63, HLA-DR1	[67]
	Human T-cell lymphotropic virus	Retroviridae	ss(+)RNA - RT	80–100 nm	ES	NS	NG	CD9, CD63, CD81, LC3, p62	[134]
	Avian leukosis virus J	Retroviridae	ss(+)RNA - RT	80–100 nm	ES	50–200 nm	NG	CD63, CD81, Tsg101	[123]
	Severe fever with thrombocytopenia syndrome virus	Phenuiviridae	segmented-ss(-)RNA	80–120 nm	ES	50–100 nm	VP	CD63, LC3	[68]
	Rift valley fever virus	Phenuiviridae	segmented-ss(-)RNA	80–120 nm	ES	50–150 nm	NG	CD63	[174]
	*Spodoptera frugiperda* ascovirus	Ascoviridae	circular dsDNA	200–400 nm	AB	5–10μm	VP	NS	[175]
	*Heliothis virescens* ascovirus 3h	Ascoviridae	circular dsDNA	200–400 nm	MV	<1000 nm	VP	NS	[176]
	Marseillevirus	Marseilleviridae	circular dsDNA	250 nm	NS	0.3–3.5μm	VP	NS	[74]
	Hepatitis B virus	Hepadnaviridae	circular dsDNA - RT	40 nm	ES	100–150 nm	NG & VP	CD63, CD81, Alix, Tsg101	[177]
	Varicela zoster virus	Herpesviridae	dsDNA	150–200 nm	ALMV	300–500 nm	VP	LC3, Rab11	[136]
	Herpes simplex virus 1 (alpha)	Herpesviridae	dsDNA	150–200 nm	ALMV	250–1000 nm	VP	CD63, CD81, LC3, Integrin β1, Flotillin-1	[137]
	Swine fever virus	Asfarviridae	dsDNA	175–215 nm	ALMV	400–800 nm	VP	BECN1, CD81, LC3	[128]

AB: apoptotic bodies; ALMV: autophagosome-like microvesicles; ES: exosomes; MV: microvesicles; NG: naked genomes; NS: not specified. VP: viral particle.

## Figures and Tables

**Figure 1 vaccines-11-01532-f001:**
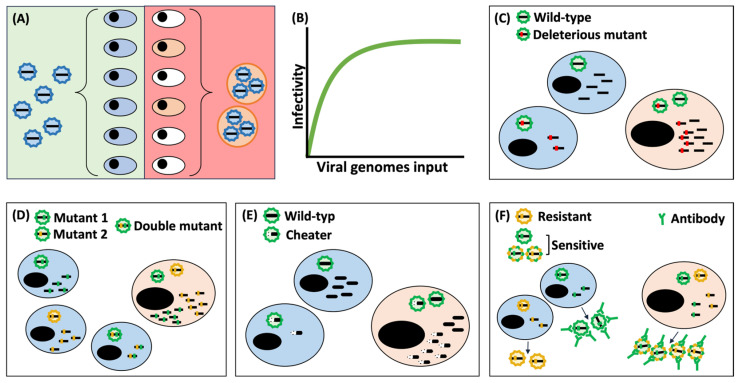
**Evolutionary implications of vesicle transmission.** (**A**) Vesicle transmission results in a reduced number of cells being reached compared to individual transmission. (**B**) The effective infection rate rises with the number of viral genomes that initiate the infection. (**C**) Within coinfected cells, trans-acting proteins carrying deleterious mutations have the potential to be complemented by functional variants, thereby restoring their fitness. (**D**) When two beneficial mutations are present in the same genome, negative epistasis can reduce the fitness of the double mutant drastically. However, co-transmission of both single mutants would allow the viral population to overcome this cost and benefit from both of them. (**E**) Within the scenario of chimeric capsids, the phenotype is primarily shaped by sensitive variants. Consequently, the co-transmission of variants in vesicles hampers the emergence of resistant variants and reinforces the efficacy of dominant drug targets. (**F**) Persistent coinfection might lead to the emergence of cheater variants, which exhibit limited replication capabilities independently but thrive in the presence of functional variants, thereby bolstering their own fitness at the expense of the population.

**Figure 2 vaccines-11-01532-f002:**
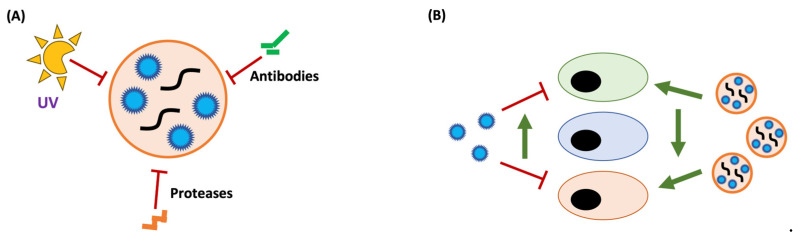
**Vesicles as a vehicle for viral transmission.** (**A**) Spreading within vesicles confers protection, effectively shielding the virus from the adverse effects of environmental conditions. (**B**) Vesicle transmission provides the virus with access to alternative routes for attachment and entry into target cells, thereby expanding the spectrum of cell types or tissues that the virus can infect.

**Table 1 vaccines-11-01532-t001:** EVs roles in viral infection.

	Function	Mechanism
Antiviral	Stimulate immune system	Transport cytokines (such as IFN or IL-6)
		Deliver PAMPs to PRRs (such as TLR)
	Damage viral RNA	Transport antiviral protein APOBEC3G
	Prevent infection	Transference of viral resistance miRNAs
	Reduce viral load	Viral receptor–viral particle binding on EV surface
	Antigen presentation	Deliver viral proteins to APCs
	Direct CD4+ and CD8+ activation	Carrying MHC–antigen complexes on EV surface
Proviral	Suppress immune system	Transport viral miRNAs/proteins, Galectin-9…
	Increase cell susceptibility	Transfer viral receptors to recipient cell’s membrane
		Promoting fusion of viral particles
		Stimulating cell–cell contacts
	Block antigen presentation	Packaging MHC complexes inside EVs
	Viral transmission	Transport infectious cargo (viral genomes and/or viral particles)

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
