# Peer review of "Trick-or-Trap: Extracellular Vesicles and Viral Transmission"

_vaccines, 2023, doi:10.3390/vaccines11101532_

Round 1

Reviewer 1 Report

Nice, well done and interesting review of a little known phenomenom which seems to potentially play a great role in  transmission of viruses. The paper is well written even if at times it is a little difficult to fully understand. Fig 1 is not very clear and could be made more clear and fully understandable.

Author Response

We thank this reviewer for the constructive comment. This reviewer is positive about the overall concept of the review but raises a point that tempers his enthusiasm. Upon reviewing the attached manuscript, you will observe substantial enhancements to its content. We have undertaken a comprehensive revision of Figure 1, now presented as “Figure 1: Evolutionary Implications of Group Transmission” and “Figure 2: Vehicle Effects of Vesicle Transmission,” designed to enrich the visual experience for our readers. Additionally, we have introduced a new, informative table titled "Table 2: EVs Roles in Viral Infection," succinctly summarizing the proviral and antiviral roles of extracellular vesicles. These changes, we believe, significantly contribute to a more profound comprehension of the concepts elucidated in our review.

Reviewer 2 Report

This manuscript explores how extracellular vesicles (EVs), lipid membrane-enclosed particles produced by cells, play a crucial role in viral transmission and infection. Unlike the traditional view of independent viral particles, viruses exploit EVs to enhance cluster transmission, increasing their chances of infecting target cells. This method offers benefits such as collective genome transmission and protection against environmental factors. Understanding EVs' role in viral transmission is vital for comprehending virus evolution and developing innovative antiviral strategies and therapies. The manuscript discusses the complexities of viral infection, barriers to success, and the concept of transmitting groups of viral genomes through various mechanisms, including EV-mediated transmission.

The manuscript would benefit from minor revisions to enhance its clarity and comprehensiveness.

The paragraph "Role of Extracellular Vesicles in Viral Infection" delves into the release of EV contents, prompting questions about the mechanisms and specificity of fusion. Further insights on these processes and their specificity could enhance the understanding (for additional details, refer to doi: 10.1155/2018/7053623).

While the manuscript highlights EVs' role in viral infection, there's an underrepresentation of their potential as antiviral agents. It would be valuable for the authors to explore broader applications of EVs and their prospects for utilization. Investigating the feasibility of modifying EVs obtained from infected cells, potentially in larger quantities than natural EVs, for use as vectors could offer exciting possibilities (doi: 10.3390/ijms231810522).

The article overlooks the potential antiviral effects of EVs sourced from healthy cells, primarily due to their contents. Addressing this point would enrich the discussion.

Moreover, the focus could extend beyond just nEVs to encompass iEVs, currently undergoing extensive research as promising therapeutic agents.

In the conclusion, the statement "Therefore, new vaccine strategies could be elaborated based on EV transmission [126]" warrants elaboration. Incorporating a few additional sentences about these innovative strategies and ongoing research would provide a comprehensive conclusion.

Author Response

We thank the reviewers for their expert insight and excellent suggestions that leads us to expand the discussion, now present in the paragraph“Clinical applications and concluding insights. Here, we provide a point-by-point summary of the changes made to the manuscript in response, with the response listed in bold

The paragraph "Role of Extracellular Vesicles in Viral Infection" delves into the release of EV contents, prompting questions about the mechanisms and specificity of fusion. Further insights on these processes and their specificity could enhance the understanding (for additional details, refer to doi: 10.1155/2018/7053623).

We find this point quite fascinating and fully agree with the suggestion. We have included a paragraph in the section 'Vehicle Effects of Extracellular Vesicles' and augmented it with references that substantiate its important relevance.

While the manuscript highlights EVs' role in viral infection, there's an underrepresentation of their potential as antiviral agents. It would be valuable for the authors to explore broader applications of EVs and their prospects for utilization. Investigating the feasibility of modifying EVs obtained from infected cells, potentially in larger quantities than natural EVs, for use as vectors could offer exciting possibilities (doi: 10.3390/ijms231810522)IF: 5.6 Q1 . The article overlooks the potential antiviral effects of EVs sourced from healthy cells, primarily due to their contents. Addressing this point would enrich the discussion. Moreover, the focus could extend beyond just nEVs to encompass iEVs, currently undergoing extensive research as promising therapeutic agents. In the conclusion, the statement "Therefore, new vaccine strategies could be elaborated based on EV transmission [126]" warrants elaboration. Incorporating a few additional sentences about these innovative strategies and ongoing research would provide a comprehensive conclusion.

As other reviewers have pointed out, we find this point quite captivating and fully concur with the suggestions. In the revised manuscript, we added a new section, where we comprehensively discuss the possibility of both natural and inducible EV for antivirals, vaccines, and clinical usage.

Reviewer 3 Report

Bou et al. provide a detailed compilation of EVs in viral infection and transmission, however, the content isn't different from the previous review article. I couldn't understand the novelty of this review article? The most important part which I would like to include on challenges to studying EVs in virus transmission and cell-to-cell communication (what new tools can be added) and how they can be manipulated for vaccine strategy using viral membrane proteins which could be of vaccines readers' interest.

Author Response

We thank the reviewer for their professional insights that inspired us to add new paragraphs. In response to the comments noted by the reviewer, we have thoughtfully restructured the final section, now bearing the title "Clinical Applications and Concluding Insights." In this section, we have seamlessly integrated pertinent information regarding the utilization of extracellular vesicles in the development of novel vaccines. Additionally, we have seized the opportunity to introduce another dimension potentially affected by group transmission, aptly detailed in the section "Pros and Cons of Viral Transmission via Extracellular Vesicles" under the header "Dominant Drug Targets."  Additionally, we have introduced a new, informative table titled "Table 2: EVs Roles in Viral Infection," succinctly summarizing the proviral and antiviral roles of extracellular vesicles.

Reviewer 4 Report

The article titled "Extracellular Vesicles as Vehicles for Viral Transmission: Pros and Cons" explores the role of extracellular vesicles (EVs) in the transmission of viruses. The author discusses various aspects of viral infection and transmission, highlighting the potential benefits and drawbacks of using EVs as vehicles for viral propagation. While the article provides valuable insights into the role of extracellular vesicles in viral transmission, there are a few potential weaknesses that should be considered:

1. Recent Developments: The article appears to rely on research and knowledge available up to a certain point in time. Given the rapid pace of scientific advancements, some recent developments or findings in the field of extracellular vesicles and viral transmission should be included.

2. Clarity of Expression: Some sections of the article could benefit from improved clarity and organization. Simplifying complex concepts and ensuring a logical flow of ideas would enhance readability and comprehension.

3. Visual Aids: The article could benefit from visual aids such as diagrams, figures, or tables to help illustrate key concepts, pathways, and interactions. Visual representations can enhance understanding, especially for readers who are not experts in the field.

 4. Practical Implications: While the article discusses the potential implications of extracellular vesicle transmission for viral evolution and antiviral strategies, it could expand further on the practical applications and potential challenges of using this knowledge in medical research and clinical settings.

Moderate editing of English language required

Author Response

We thank the reviewers for their professional insights that inspired us to discuss a new Topic. We take the reviewers' comments seriously and have included several latest research findings. In particular, we have added a new section discussing research in the area of clinical applications as noted in comment 4. We hope that these additional statements make the interpretation of this review clear.

  1. Recent Developments: The article appears to rely on research and knowledge available up to a certain point in time. Given the rapid pace of scientific advancements, some recent developments or findings in the field of extracellular vesicles and viral transmission should be included.

As noted, the Manuscript Script has been revised to cite new and recent papers in each section.

  1. Clarity of Expression: Some sections of the article could benefit from improved clarity and organization. Simplifying complex concepts and ensuring a logical flow of ideas would enhance readability and comprehension.

We fully agree with this comment. Especially in the paragraphs related to Figure 1, we have considered the structure of the paper and created new figures and tables. 3.

  1. Visual Aids: The article could benefit from visual aids such as diagrams, figures, or tables to help illustrate key concepts, pathways, and interactions. Visual representations can enhance understanding, especially for readers who are not experts in the field.

I also agree with this comment. A new table has been created and revised to provide a comprehensive understanding.

  1. Practical Implications: While the article discusses the potential implications of extracellular vesicle transmission for viral evolution and antiviral strategies, it could expand further on the practical applications and potential challenges of using this knowledge in medical research and clinical settings.

As other reviewers have pointed out, we find this point quite captivating and fully concur with the suggestions. In the revised manuscript, we added a new section, where we comprehensively discuss the possibility of both natural and inducible EV for antivirals, vaccines, and clinical usage.

Round 2

Reviewer 4 Report

The manuscript has been revised by the authors in accordance with the reviewer's comments, making it suitable for publication in Vaccines.

Minor editing of English language required